# A Temperature Drift Suppression Method of Mode-Matched MEMS Gyroscope Based on a Combination of Mode Reversal and Multiple Regression

**DOI:** 10.3390/mi13101557

**Published:** 2022-09-20

**Authors:** Liangqian Chen, Tongqiao Miao, Qingsong Li, Peng Wang, Xuezhong Wu, Xiang Xi, Dingbang Xiao

**Affiliations:** College of Intelligence Science and Technology, National University of Defense Technology, Changsha 410073, China

**Keywords:** MEMS gyroscope, mode reversal, temperature drift suppression

## Abstract

In recent years, the application prospects of high-precision MEMS gyroscopes have been shown to be very broad, but the large temperature drift of MEMS gyroscopes limits their application in complex temperature environments. In response to this, we propose a method that combines mode reversal and real-time multiple regression compensation to compensate for the temperature drift of gyroscope bias. This method has strong adaptability to the environment, low computational cost, the algorithm is online in real time, and the compensation effect is good. The experimental results show that under the temperature cycle of −20~20 °C and the temperature change rate of 4 °C/min, the method proposed in this paper can reduce the zero-bias stability from about 27.8°/h to 0.4527°/h, and the zero-bias variation is reduced from 65.88°/h to 1.43°/h. This method improves the zero-bias stability of the gyroscope 61-fold and the zero-bias variation 46-fold. Further, the method can effectively suppress the zero-bias drift caused by the heating of the gyroscope during the start-up phase of the gyroscope. The zero-bias stability of the gyroscope can reach 0.0697°/h within 45 min of starting up, and the zero-bias repeatability from 0 to 5 min after startup is reduced from 0.629°/h to 0.095°/h.

## 1. Introduction

In recent years, with the continuous development of high-precision microelectromechanical systems (MEMS) gyroscope performance and processing technology, MEMS gyroscope has become an important part of autonomous navigation [1,2], unmanned systems, and precision guidance due to its low cost, low power consumption, and small size [3,4,5]. As a sensor, the zero-bias performance of a MEMS gyroscope is an important criterion for evaluating the quality of a gyroscope. However, the characteristic of MEMS gyroscope’s zero-bias drift with temperature changes makes its working performance under the conditions of full temperature and rapid temperature change become very poor, thus limiting its application on many occasions with large temperature changes or high-performance requirements.

In response to this problem, many scholars have carried out the work of compensating for the zero-bias temperature drift of MEMS gyroscopes. There are many ways to solve this problem: one is to optimize the design of the gyroscope, which usually includes the optimization of the structure and circuit. For example, Zhanshe Guo [6] designed a resonant MEMS gyroscope with a temperature self-compensation function; Huiliang Cao [7,8,9] proposed a new method for MEMS gyroscope temperature compensation for peripheral circuits. Another similar method is on-chip temperature control. This method uses a temperature sensor for temperature measurement, and a heating platform for temperature control, so that the gyroscope works at a constant temperature [10,11]. Although this method can theoretically allow the gyroscope to operate free from ambient temperature interference, its implementation requires a complex on-chip manufacturing process and consumes more power. Another way is to compensate for the zero-bias temperature drift of the gyroscope, which is usually a more efficient method. At first, the temperature information provided by the external temperature sensor was used to compensate for the temperature drift of the gyroscope, but the external temperature is often inconsistent with the operating temperature inside the gyroscope, so there will be thermal hysteresis [12]. This reduces the compensation effect of this method when the temperature changes sharply or irregularly and the gyroscope itself heats up obviously. Therefore, subsequently, the driving frequency of gyroscopes [13] or some other real-time output of gyroscopes was used as a measure of temperature to compensate for the temperature drift. The compensation methods used are also different. The most common way is to use polynomials [14] for compensation. This method is simple and requires little calculation, but it is usually based on only one variable, and the compensation effect is poor for different temperature change rates or rapid temperature changes. Therefore, researchers use the method of multi-parameter multiple regression algorithm [15]. Since there are multiple variables for comprehensive consideration, the inaccurate compensation caused by the inaccurate correspondence of a certain variable can be reduced. In addition, some compensation methods based on neural network genetic algorithm [16,17,18] and machine-learning techniques [19] are adaptable to the environment, but they often require a large amount of data to train them, which requires a large amount of calculation. In addition, there are many researchers who combine the above methods with some optimization algorithms [20,21,22] such as the Kalman filter [23], WLMP, and CS-SVR [24] to comprehensively compensate for them, and this approach has produced some results.

Mode reversal is a real-time online self-compensation method for gyroscope bias, which can eliminate most of the bias drift errors of the gyroscope structure. This method significantly improves the turn-on repeatability and zero-bias stability of the gyroscope in the variable temperature range [25], and can effectively suppress the long-term temperature drift of the gyroscope [26]. Based on the zero-bias drift mechanism, the method does not require additional sensors and complicated manufacturing processes, and can realize self-calibration in real-time operation. However, with this method, it is difficult to suppress the zero-bias temperature drift introduced by the detection method of the measurement and control circuit and some electronic components [27]. Meanwhile, under the condition of rapid temperature change, due to the temperature difference caused by the time difference between the two modes of operation, mode reversal has a limited effect on the suppression of temperature drift.

Aiming at the above problems, this paper studies the zero-bias temperature drift compensation of the honeycomb disk resonator gyroscope (HDRG) [28,29,30]. The temperature drift is compensated for by using the mode reversal to suppress the zero-bias drift combined with the multiple regression algorithm. This method not only solves the remaining zero-bias drift caused by the transition time of the mode reversal under the condition of rapid temperature change by compensating for the zero-bias temperature drift with the multiple regression algorithm, but also uses the mode reversal to suppress the zero-bias drift, which makes the compensation effect of the multiple regression algorithm better. The rest of the paper is structured as follows: Section 2 introduces the HDRG used in this experiment. Section 3 introduces the zero-bias drift principle of the cellular gyroscope and the principle of mode reversal to suppress the zero-bias drift, and outlines the zero-bias temperature compensation method we use. Section 4 conducts temperature compensation experiments and compares the effects of different compensation methods to verify the superiority of this method. Finally, conclusions are given in Section 5.

## 2. Honeycomb Disk Resonator Gyroscope

The honeycomb disk resonator gyroscope studied in this paper is mainly composed of a resonant structure and electrodes, as shown in Figure 1a. The resonant structure includes two parts, an anchor point and a frame structure, wherein the anchor point is fixedly connected with the base and plays a supporting role, and the frame structure is suspended in the air and plays the role of sensitive angular velocity. The electrodes are fixedly connected to the substrate through anchor points, including electrodes outside the resonance frame (outer electrodes) and electrodes inside the resonance frame (inner electrodes). An equivalent parallel plate capacitance is formed between the electrode and the adjacent resonant structure. The vibration of the gyroscope can be sensed by detecting changes in the capacitance.

The HDRG has various vibration modes. Since the driving electrodes and driving force are located in the plane (the plane where the diameter of the gyroscope is located), the in-plane mode is the main vibration mode. According to the difference in the number of antinodes n, the mode of the gyroscope can be defined as n=1, 2, 3… order modes. The honeycomb disk resonator gyroscope in this paper works in the degenerate mode of n=2. The schematic diagram of the mode is shown in Figure 1c.

The SEM image of the HDRG is shown in Figure 1b. The MEMS manufacturing processes of the HDRG include anchors etching, thermal oxidation, wafer fusion bonding, polishing, Al pads patterning, and deep reactive ion etching (DRIE) process. The detailed process flow for the preparation of the HDRG is reported in a previous paper [29].

In this paper, a modal test circuit and a lock-in amplifier are used to test the frequency response and decay curve of the honeycomb resonator in order to characterize the basic performance parameters such as resonant frequency, quality factor, and decay time constant of its driving mode and sensing mode. The test results are shown in Figure 1d,e.

## 3. Zero-Bias Temperature Drift Compensation Method

### 3.1. Zero-Bias Drift Principle

The zero-bias performance of the gyroscope is mainly determined by the noise of the gyroscope output signal and the zero-bias drift. For the HDRG, the main sources of noise are external noise caused by external circuits or natural factors, and inherent noise composed of mechanical thermal noise, high- and low-pass op-amp noise, DC bias voltage noise, etc. The zero-bias drift includes the drift of the gyroscope structure parameters and the drift of the measurement and control circuit. The drift of the circuit cannot be accurately estimated due to the circuit layout design, selection of electronic components, and other issues, so here we only discuss the zero-bias drift caused by the parameter change in the gyroscope structure itself.

The zero-bias drift of the gyroscope structure itself may come from the temperature change of the external environment, the release of structural stress, the aging of the gyroscope, and the change in the heat distribution during the vibration of the gyroscope itself. Among them, structural stress release and gyroscope aging can speed up the change process by long-time annealing after processing, and the effects of these two drifts can be ignored in the annealed gyroscope. During the vibration of the gyroscope itself, heat flows due to the tensile and compressive deformation of the elastic beam. The heat flow may cause changes in the material properties such as the elastic modulus and density of the gyroscope, which in turn lead to changes in the damping axis declination $\theta_{\tau }$ of the gyroscope. The temperature change of the external environment is the main source of bias drift, and the frequency, damping, and damping axis deflection angle $\theta_{\tau }$ caused by temperature change will directly affect the gyroscope bias stability. For the HDRG, the mathematical model of its zero-bias drift can be expressed as:(1)$$\begin{array}{c} \partial \Omega_{Bias}=\frac{1}{2nA_{g}}\left[\partial \left(\frac{2}{\tau_{1}}\alpha_{1}\right)-\partial \left(\left(\Delta \frac{1}{\tau }\right)\sin2\theta_{\tau }\right)\right]=\frac{1}{2nA_{g}}[\alpha_{1}\left(\frac{\partial \omega_{1}}{Q_{1}}-\partial Q_{1}\frac{\omega_{1}}{Q_{1}{}^{2}}\right)-\\ \partial \theta_{\tau }\left(\frac{\omega_{1}}{Q_{1}}-\frac{\omega_{2}}{Q_{2}}\right)\cos2\theta_{\tau }-\frac{1}{2}\sin2\theta_{\tau }\left(\frac{\partial \omega_{1}}{Q_{1}}-\partial Q_{1}\frac{\omega_{1}}{Q_{1}{}^{2}}-\frac{\partial \omega_{2}}{Q_{2}}+\partial Q_{2}\frac{\omega_{2}}{Q_{2}{}^{2}}\right)] \end{array}$$
where $\omega_{1}$, $\omega_{2}$, $Q_{1}$, and $Q_{2}$ are the frequency and quality factor at room temperature, n is gyroscope mode number, $A_{g}$ is angle gain, and $\alpha_{1}$ is driving angle.

By measuring the frequency, quality factor, and the variation law of the damping axis deflection angle of the gyroscope with temperature, the zero-bias stability performance of the gyroscope can be approximately estimated, but because the drift of the circuit also occupies a certain component, the above formula can only be used for preliminary estimation of the performance of the gyroscope.

### 3.2. Temperature Self-Sensing Based on the Frequency of the Gyroscope

First of all, we can discuss the effect of temperature on the frequency of the gyroscope. The resonant frequency of the gyroscope is related to the structural parameters and material properties of the gyroscope. For the HDRG, we can express the frequency of the gyroscope as:(2)$$f=\vartheta h\sqrt{\frac{E}{\rho }}$$
where ϑ is the related quantity of the gyroscope structure, h is the wall thickness of the gyroscope ring and spokes, and ρ is the density, both of which are related to the thermal expansion coefficient α of the material, and the thermal expansion coefficient of single crystal silicon is about 2.6 ppm/°C. E is the elastic modulus. Since the elastic modulus of single crystal silicon is different [31] in each direction, the temperature coefficient of the elastic modulus in each direction is also different. In addition, the doping rate of the silicon elastic modulus material is also closely related. Research has shown that the change in elastic modulus of silicon is approximately linear at room temperature of 20 °C (293.15 K), and its temperature coefficient $k_{E}$ is about −60 ppm/°C. Therefore, the resonant frequency can be further expressed as:(3)$$\begin{array}{c} f\left(\Delta T\right)=\vartheta h_{0}\left(1+\alpha \Delta T\right)\sqrt{\frac{E_{0}\left(1+k_{E}\Delta T\right)}{\rho_{0}\left(1-3\alpha \Delta T\right)}}\approx \vartheta h_{0}\sqrt{\frac{E_{0}}{\rho_{0}}}\left(1+\frac{k_{E}+5\alpha }{2}\Delta T\right)\\ =f_{0}\left(1+k_{\omega }\Delta T\right) \end{array}$$

It can be seen from this formula that the frequency of the HDRG has a linear relationship with the temperature within a certain range within the linear vibration range. Therefore, we tested the honeycomb disk resonator gyroscope prototype, and obtained the change in the resonant frequency of the driving shaft of the gyroscope with temperature as shown in the Figure 2. It can be seen that the change in the HDRG with temperature is a linear law. Taking 0 °C as the benchmark, the temperature coefficient of the resonant frequency is about −20.9 ppm.

According to the experimental results in Figure 2, it can be seen that the driving frequency of the HDRG has a linear relationship with the temperature change. And the driving frequency of the gyroscope is the real-time output of the gyroscope, so we can use the driving frequency as the reference for the temperature change to compensate for the temperature drift of the zero bias of the gyroscope.

### 3.3. Suppression of Zero Bias Drift by Mode Reversal

Mode reversal is a real-time online self-compensation method for gyroscope bias by rotating the gyroscope mode angle by 90°. We assume that x and y are the driving axis and sensing axis of the HDRG. Here we consider the damping shaft deflection angle $\theta_{\tau }$ as the main source of bias drift. As shown in Figure 3a, for the HDRG, if the driving axis is changed from x to y and the sensing axis is changed from y to x, the zero bias of the gyroscope will be:(4)$${\;\Omega }_{x}=\frac{1}{2nA_{g}}\left[\left(\frac{2}{\tau_{1}}\alpha_{1}\right)-\left(\Delta \frac{1}{\tau }\right)\sin2\theta_{\tau }\right]\;\Omega_{y}=\frac{1}{2nA_{g}}\left[\left(\frac{2}{\tau_{2}}\alpha_{2}\right)+\left(\Delta \frac{1}{\tau }\right)\sin2\theta_{\tau }\right]$$

Adding the two zero biases gives:(5)$${\;\Omega }_{x}+\Omega_{y}=\frac{1}{2nA_{g}}\left(\frac{2}{\tau_{1}}\alpha_{1}+\frac{2}{\tau_{2}}\alpha_{2}\right)\approx 0$$
where $\alpha_{1}$ and $\alpha_{2}$ are driving angle, which is close to 0°. It can be seen that the superposition of the zero-bias output after the axis change can eliminate the influence of the damping asymmetric term on the zero bias and its drift, which is the principle of mode reversal [32,33] to suppress the zero-bias drift. In this way, we can eliminate the effect of the damping shaft deflection angle $\theta_{\tau }$ on the zero-bias drift.

The electronic and control scheme of mode reversal in the HDRG is shown in Figure 3c. The automatic gain control (AGC) loop is used for exciting driving mode with constant amplitude, while the force-to-rebalance (FTR) loop is designed for detecting angular velocity by solving the rebalanced force in the sensing mode. In addition, virtual switches (VSs) are placed in both the AGC and FTR loop of the HDRG. This way, we can control the drive angle α by opening and closing the VSs on the FPGA.

### 3.4. Real-Time Rapid Temperature Change Compensation Scheme in Mode Reversal

It can be seen from the derivation in the previous section that mode reversal can eliminate the influence of the damping asymmetric term on the zero bias and its drift, thereby eliminating a large part of the zero-bias drift. The working sequence of mode reversal is shown in Figure 3b. Mode 1 and mode 2 are periodically exchanged to realize self-compensation for zero-bias drift. In fact, in the case of slow temperature changes, the mode reversal can well suppress the zero-bias drift. However, in the case of rapid temperature change, the temperature change caused by the working time difference between mode 1 and mode 2 is obvious, so the mode reversal cannot well suppress the zero-bias drift. Similarly, for the drift caused by residual stress, parasitic charge accumulation [34,35], electronic component, and the measurement and control circuit, we can define it as $B_{E}$, which is related to temperature. Then, for the zero-bias drift of the gyroscope working in mode reversal, we can express it as:(6)$$B_{drift}=\partial \Omega_{x\_T1}+\partial \Omega_{y\_T2}+B_{E}\propto f\left(T\right)+f\left(\frac{dT}{dt}\right)+f\left(T\cdot \frac{dT}{dt}\right)$$
where $\Omega_{x\_T1}$ means the zero bias of the HDRG working in mode 1 when the operating temperature is T1, $\Omega_{y\_T2}$ means the zero bias of the HDRG working in mode 2 when the operating temperature is $T2;\;f\left(T\right)$ is a polynomial of frequency, and $f\left(\frac{dT}{dt}\right)$ is a polynomial of the frequency change rate. $f\left(T\Delta \frac{dT}{dt}\right)$ is the coupling term of each order of the two independent variables. To this end, we propose a way to combine multiple regression compensation with mode reversal to compensate for zero-bias drift.

## 4. Temperature Compensation Experiments

### 4.1. The Experimental Setup

To verify the feasibility and effectiveness of our method, we tested the temperature characteristics of the gyroscope through temperature experiments. The different types of experiments are given in the following subsections. The experimental platform we used is shown in Figure 4. The main experimental equipment used is a temperature and humidity test chamber, which is used to create different temperature environments; a DC power supply, which is used to power and ground the HDRG; an upper computer, which is used to collect experimental data and control the working state of the HDRG.

### 4.2. Polynomial Compensation in Slowly Varying Temperature Condition

In the case of a slow rate of temperature change, we can first try to compensate for the drift of the zero bias with temperature using a polynomial. Assuming that the zero bias of the gyroscope is y, the driving frequency is x, and the measured value is $\left(x_{i},y_{i}\right)$, the compensation model can be established as follows:(7)$$\begin{array}{c} y=a_{0}+a_{1}x+a_{2}x^{2}+\cdots +a_{m}x^{m}\\ y_{i}=a_{0}+a_{1}x_{i}+a_{2}x_{i}{}^{2}+\cdots +a_{m}x_{i}{}^{m}+v_{i}\;,\;i=1,2,\cdots ,n\\ R=\sum v_{i}{}^{2} \end{array}$$
where $a_{0},a_{1},\cdots ,a_{m}$ is the coefficients of each term, $v_{i}$ is the residual of the model. R is the residual sum of squares of the model, which should be regressed to a minimum. We can first explore the degree of the polynomial, and determine the optimal degree of the polynomial according to the fitting effect. Compensation is performed on a set of zero bias data whose temperature slowly rises from −40 °C to 60 °C, and then drops to 25 °C after heat preservation. Polynomials of different orders are used to compare the compensation effects. The results are shown in Table 1. Considering the compensation effect and the computational cost of subsequent algorithms together, we think that quadratic or cubic polynomials are more appropriate. The data and compensation results are shown in Figure 5a. Figure 5b is the data and polynomial fitting results of the corresponding relationship between driving frequency and zero bias.

### 4.3. Multiple Regression Compensation under Temperature Cycling Condition

The compensation method in the previous section is only suitable for the temperature drift compensation of the zero bias with a slow and constant temperature change rate. So, it just serves as a reference for us to determine the order of the compensation model. In fact, the zero-bias drift of the gyroscope is also related to the temperature change rate, which is why the zero-bias of the gyroscope will have a hysteresis phenomenon in the temperature cycle experiment. Therefore, we introduce the temperature change rate and the coupling terms of temperature and temperature change rate, and re-establish a multiple regression compensation model.
(8)$$\begin{array}{c} B_{0}=k_{0}+k_{1\sim 3}\cdot f\left(T\right)+k_{4\sim 6}\cdot f\left(\frac{dT}{dt}\right)+k_{8\sim 16}\cdot f\left(T\cdot \frac{dT}{dt}\right)\\ B_{0}{}_{i}=k_{0}+k_{1\sim 3}\cdot f\left(T_{i}\right)+k_{4\sim 6}\cdot f\left(\frac{dT_{i}}{dt}\right)+k_{8\sim 16}\cdot f\left(T_{i}\cdot \frac{dT_{i}}{dt}\right)+v_{i}\;,\;i=1,2,\cdots ,n\\ k_{1\sim 3}\cdot f\left(T\right)=k_{1}\cdot T+k_{2}\cdot T^{2}+k_{3}\cdot T^{3},\;k_{4\sim 6}\cdot f\left(\frac{dT}{dt}\right)=k_{4}\cdot \frac{dT}{dt}+k_{5}\cdot {\left(\frac{dT}{dt}\right)}^{2}+k_{6}\cdot {\left(\frac{dT}{dt}\right)}^{3}\\ k_{8\sim 16}\cdot f\left(T\cdot \frac{dT}{dt}\right)=k_{8}\cdot T\cdot \frac{dT}{dt}+k_{9}\cdot T\cdot {\left(\frac{dT}{dt}\right)}^{2}+k_{10}\cdot T\cdot {\left(\frac{dT}{dt}\right)}^{3}+k_{11}\cdot T^{2}\cdot \frac{dT}{dt}+k_{12}\cdot T^{2}\\ \cdot {\left(\frac{dT}{dt}\right)}^{2}+k_{13}\cdot T^{2}\cdot {\left(\frac{dT}{dt}\right)}^{3}+k_{14}\cdot T^{3}\cdot \frac{dT}{dt}+k_{15}\cdot T^{3}\cdot {\left(\frac{dT}{dt}\right)}^{2}+k_{16}\cdot T^{3}\cdot {\left(\frac{dT}{dt}\right)}^{3}\\ R=\sum v_{i}{}^{2} \end{array}$$
where, $B_{0}$ is the gyroscope bias, $f\left(T\right)$ is a cubic polynomial of frequency, and $f\left(\frac{dT}{dt}\right)$ is a cubic polynomial of the frequency change rate; $f\left(T\cdot \frac{dT}{dt}\right)$ is the coupling term of each order within the third order of the two independent variables, and $k_{0}\sim k_{16}$ are the regression coefficients, $v_{i}$ is the residual of the model. R is the residual sum of squares of the model, which should be regressed to a minimum. For our gyroscope with a sampling rate of 1 Hz, the frequency change rate $\frac{dT}{dt}=\frac{\Delta T}{\Delta t}=\frac{T_{2}-T_{1}}{t_{2}-t_{1}}=T_{2}-T_{1}$ can be approximately expressed as a frequency difference of two seconds, which is also real-time output. In fact, this is a generalized model that can be applied to all HDRGs. Since the temperature characteristics of each HDRG are not the same, the compensation equations they regress are also different. For a particular HDRG, it may not use every term in the compensation model. Therefore, in practical applications, we can simplify the model according to the actual situation.

We put the gyroscope in the temperature and humidity test chamber to perform a temperature cycle process of −20~20 °C, and set different temperature change rates for the temperature box to obtain the zero-bias drift data of the gyroscope. We use the compensation model given above to compensate the temperature cycle data with different temperature change rates, and the obtained compensation result curve is shown in Figure 6. Figure 7 shows the relationship between the drive frequency and the zero bias when the temperature box is set with different temperature changing rates. Table 2 shows the comparison of the compensation effect of each group of data self-compensation. From Figure 6 and Figure 7 and Table 2, we can see that the compensation model has a good compensation effect in the face of different temperature change rates, and it is proved that the introduction of the temperature change rate is necessary for the zero-bias drift compensation of the gyroscope.

Then, we select data 1 as the benchmark, determine the various coefficients of the compensation model, and then use this determined model to compensate other data; the obtained compensation curve is shown in Figure 8. Table 3 shows the compensation effect of using data 1 as a benchmark to compensate for other data. Compared with the above self-compensation, the loss of compensation effect is not too large, the zero-bias stability and the zero-bias variation can be improved by about an order of magnitude, and the zero-bias repeatability can reach 1.5°/h.

### 4.4. Rapid Temperature Change Compensation in Mode Reversal

In the previous section we validated our compensation method for multiple regression. The HDRG working in mode reversal can also effectively suppress the zero-bias drift [36]. However, there is still some residual bias drift after mode reversal under the condition of rapid temperature change, which is shown in Formula (6). We use the multiple regression compensation described in the previous section combined with mode reversal, so that the temperature drift of the zero bias is suppressed to a very small extent.

Similarly, in the temperature cycle of −20~20 °C and the temperature change rate of the temperature and humidity test chamber at 4 °C/min, the mode reversal period is set to 300 s. The mode reversal period is 300 s, which means that the gyroscope needs to perform two mode exchanges in one cycle, and each mode reversal time is about 30 s. As shown in Figure 2b, for the working sequence of one cycle of the single gyroscope mode reversal, the gyroscope first performs the exchange time of 30 s, then uses mode 1 to work for 120 s, then performs the mode reversal for 30 s, and finally uses mode 2 to work for 120 s. That is, about 180 s after the gyroscope is turned on, the gyroscope output of the mode reversal can be obtained. The zero-bias stability of the gyroscope under mode reversal is reduced from about 27.8°/h to 2.5653°/h, and the zero-bias variation is reduced from 65.88°/h to 6.12°/h. The multiple regression compensation described in the previous section is simultaneously performed on the gyroscope working in mode reversal, and the compensation result is shown in Figure 9. Using the compensation method of mode reversal combined with multiple regression, the zero-bias stability can be reduced from about 27.8°/h to 0.4527°/h, and the zero bias variation can be reduced from 65.88°/h to 1.4308°/h. In this way, the zero-bias stability of the gyroscope is increased 61-fold, the zero-bias range is increased 46-fold, and the zero-bias drift of the gyroscope with temperature is greatly suppressed.

The drift caused by the turn-on process of the gyroscope [37,38,39,40] is considered to be the result of parasitic charge accumulation and the rapid heating process after power-on. This process fits well with the situation described in Formula (6). The drift caused by the rapid heating process is $\partial \Omega_{x\_T1}+\partial \Omega_{y\_T2}$, and the drift caused by parasitic charge accumulation is $B_{E}$. In order to verify the compensation effect in mode reversal in practice, we compensate for the zero-bias drift caused by the rapid heating of the gyroscope during the turn-on process of the HDRG. We took the data of the gyroscope being turned on for 45 min; here we define 40 min after booting as the unstable stage of startup, and 40 min later as the completely stable stage after booting. In order to highlight only the zero-bias drift, we used the average zero bias from 40 min to 45 min of start-up as the offset baseline, eliminating the effect of the zero-bias offset. The compensation results are shown in Figure 10, and the comparison between the compensation results is shown in Table 4. After four sets of data compensation, the zero-bias stability of the gyroscope can reach an average level of 0.0697°/h. It is comparable to the zero-bias stability level of the gyroscope after stable operation at room temperature, and the drift of the zero-bias with temperature is basically compensated for. In addition, the zero-bias repeatability from 0 to 5 min of turn-on is reduced from 0.629°/h to 0.095°/h. Basically, the influence of boot time on gyroscope performance is eliminated.

## 5. Conclusions

In this paper, a method combining mode reversal and real-time multiple regression compensation is proposed to compensate for the temperature drift of the HDRG. The multivariate regression algorithm uses the gyroscope driving frequency, the driving frequency variation, and the coupling term of the two as independent variables to sense the temperature change, thereby compensating for the temperature drift of the zero bias. The experimental results show that using this multiple regression algorithm alone under the temperature cycle condition of −20~20 °C can improve the zero-bias stability and zero-bias variation of the gyroscope by an order of magnitude under different temperature change rates. The mode reversal can suppress the offset drift by eliminating the influence of the gyroscope damping axis deflection angle on the offset. Under the conditions of −20~20 °C temperature cycle and 4 °C/min temperature change rate of the incubator, the gyroscope in mode reversal can also improve the zero-bias stability and zero-bias variation by about an order of magnitude. Under the same conditions, using the method proposed in this paper combined with mode reversal and real-time multiple regression compensation, the zero-bias stability can be reduced from about 27.8°/h to 0.4527°/h, and the zero-bias variation can be reduced from 65.88°/h to 1.43°/h. The zero-bias stability of the gyroscope is increased 61-fold, and the zero-bias variation is increased 46-fold. Finally, we apply this method to the gyroscope to compensate for the zero-bias drift caused by the rapid heating of the gyroscope during the startup phase of the gyroscope. The zero-bias stability of multiple sets of data within 45 min of starting up reaches 0.0697°/h, which basically compensates for the temperature drift of the zero bias during the start-up phase of the gyroscope. In addition, the zero-bias repeatability from 0 to 5 min of turn-on is reduced from 0.629°/h to 0.095°/h, which basically eliminates the effect of start-up time on gyroscope performance.

## Figures and Tables

**Figure 1 micromachines-13-01557-f001:**
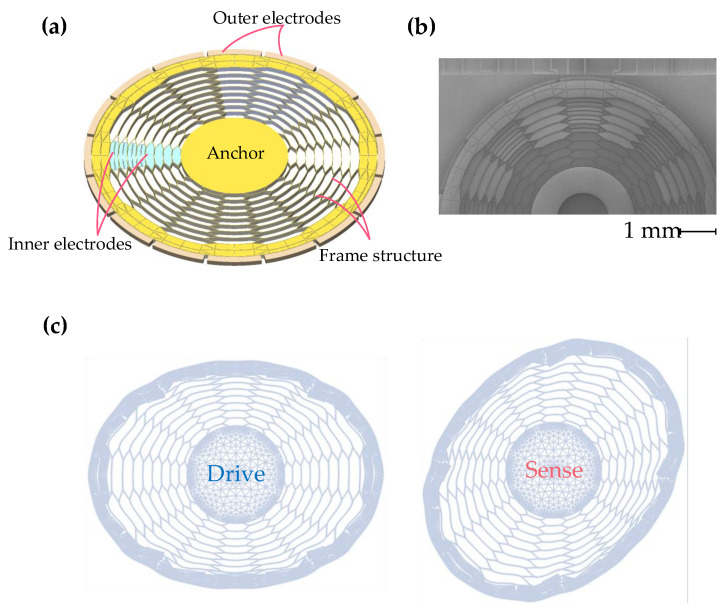
(**a**) Honeycomb disk resonator gyroscope structure. (**b**) Schematic diagram of the mode. (**c**) SEM image of the HDRG. (**d**) Resonant frequency and initial frequency difference of driving mode and sensing mode. (**e**) Ring-down curves of the driving and sensing modes of the HDRG.

**Figure 2 micromachines-13-01557-f002:**
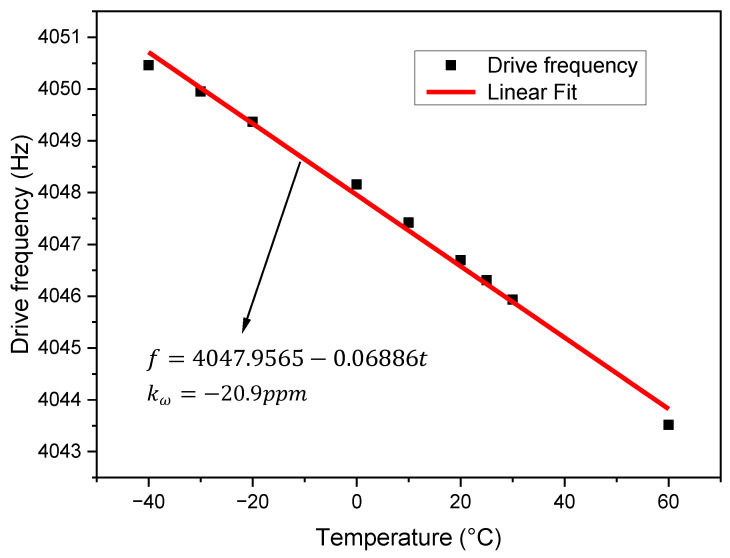
Influence of temperature change on the resonant frequency of the HDRG.

**Figure 3 micromachines-13-01557-f003:**
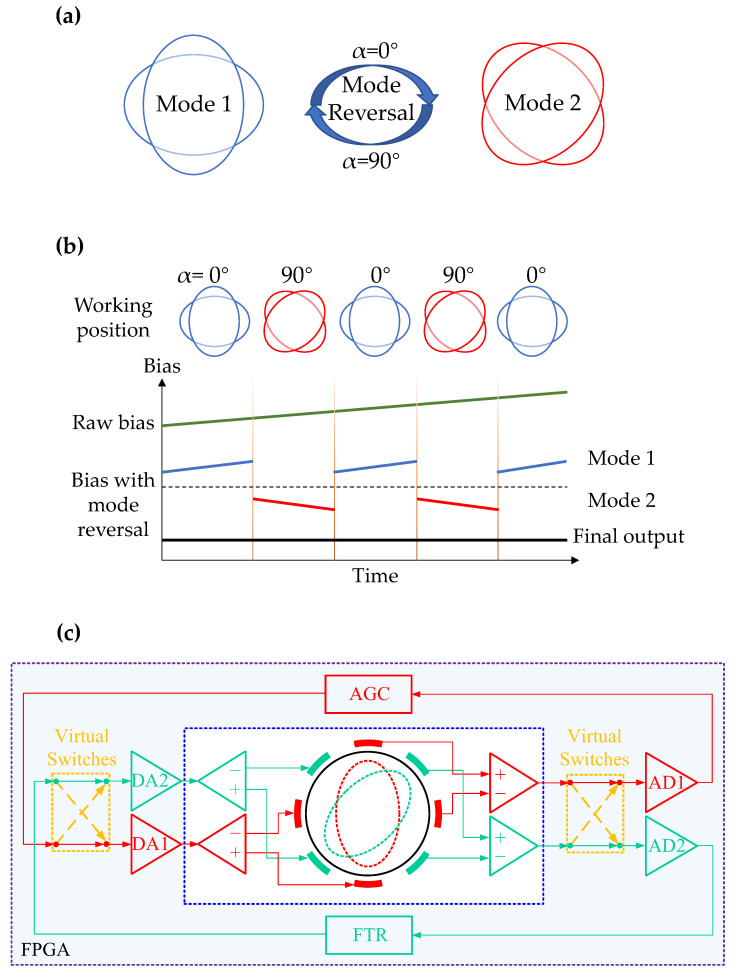
(**a**) Mode reversal schematic. (**b**) Mode reversal working sequence diagram. (**c**) Electronic and control scheme of mode reversal in the HDRG.

**Figure 4 micromachines-13-01557-f004:**
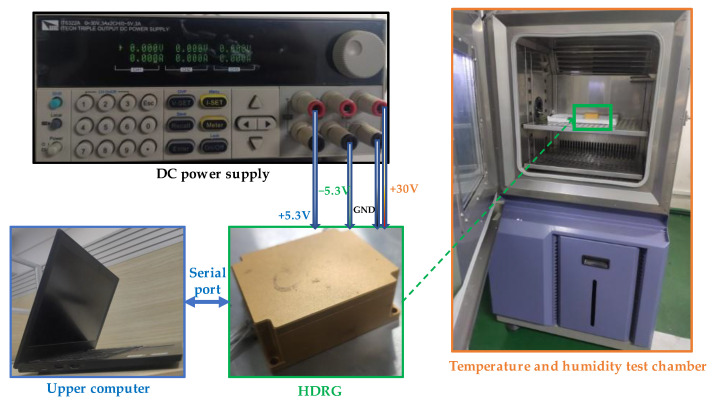
Experimental platform and main equipment.

**Figure 5 micromachines-13-01557-f005:**
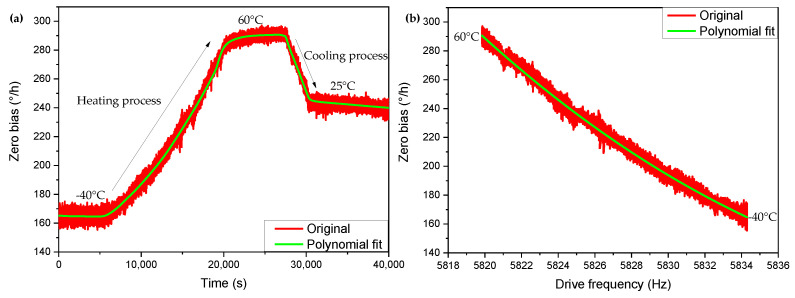
Zero-bias drift data under slowly changing temperature conditions, polynomial fit, and zero bias after compensation. (**a**) Zero-bias time curve. (**b**) Zero bias-drive frequency curve.

**Figure 6 micromachines-13-01557-f006:**
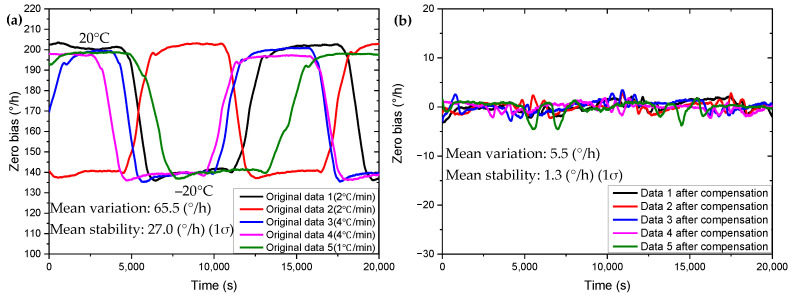
Zero-bias drift self-compensation results under different temperature change rates. (**a**) Before compensation. (**b**) After compensation.

**Figure 7 micromachines-13-01557-f007:**
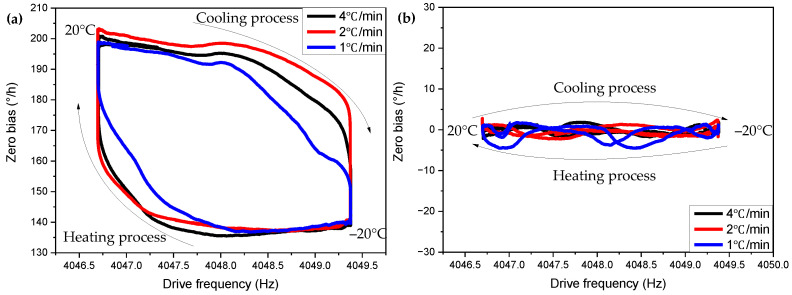
Relationship between gyroscope driving frequency and zero bias under different temperature change rate conditions. (**a**) Before compensation. (**b**) After compensation.

**Figure 8 micromachines-13-01557-f008:**
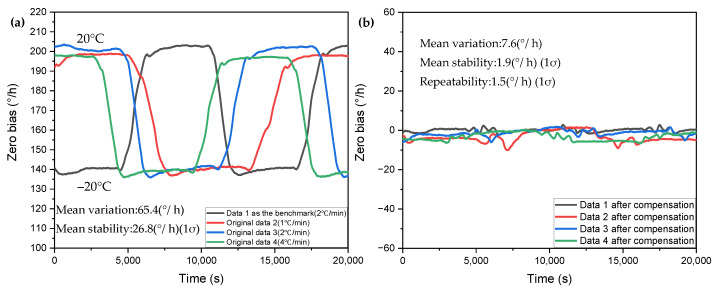
Determine the regression equation with data 1, and use the regression equation to compensate for the compensation results of other data. (**a**) Before compensation. (**b**) After compensation.

**Figure 9 micromachines-13-01557-f009:**
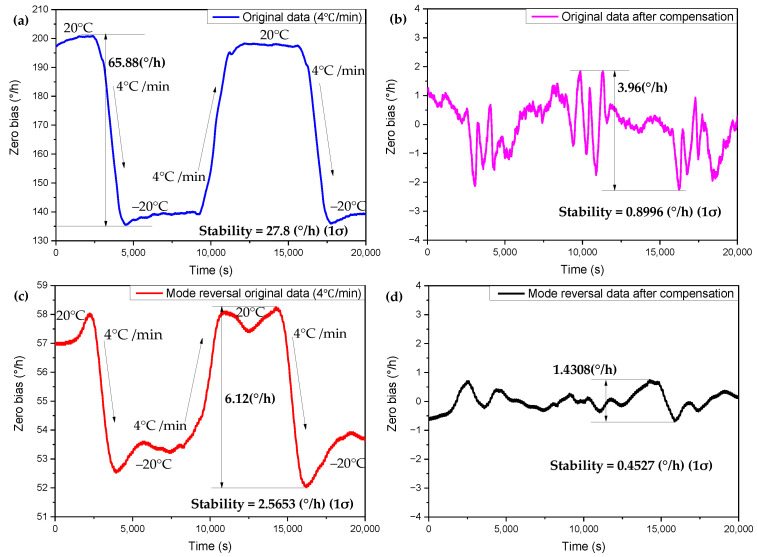
−20~20 °C temperature cycle, under the condition of 4 °C/min temperature change rate, the zero bias of the mode reversal, and the zero-bias comparison of the mode reversal combined with multiple regression compensation. (**a**) Working in FTR mode before compensation. (**b**) Working in FTR mode after compensation. (**c**) Working in mode reversal before compensation. (**d**) Working in mode reversal after compensation.

**Figure 10 micromachines-13-01557-f010:**
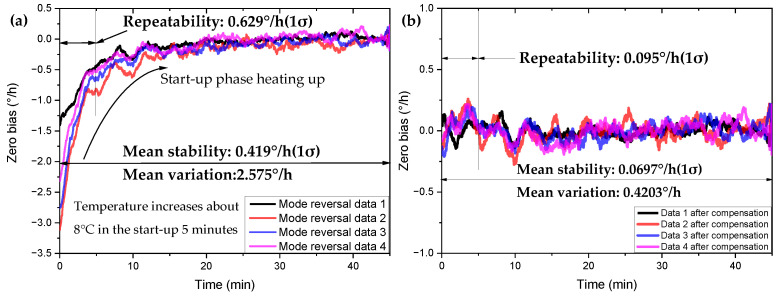
Comparison of zero-bias drift compensation results in the rapid heating-up stage of the gyroscope in mode reversal. (**a**) Before compensation. (**b**) After compensation.

**Table 1 micromachines-13-01557-t001:** Comparison of fitting effects of different polynomial degrees.

Type of Data Processing	Zero-Bias Variation (°/h)	Zero-Bias Stability (°/h)	Zero-Bias Mean (°/h)
original data	129.4627	15.7944	-
First-order polynomial compensation	18.7559	3.9337	$-3.5175e^{-11}$
Quadratic polynomial compensation	13.1759	2.4561	$-4.8545e^{-9}$
Cubic polynomial compensation	13.1440	2.4649	$5.9379e^{-7}$
Quartic polynomial compensation	13.9183	2.3488	0.0024
Quintic polynomial compensation	13.9168	2.3489	0.0053
Sixth degree polynomial compensation	14.2046	2.3323	0.5081
Seventh degree polynomial compensation	14.6513	2.3204	0.1665
Eighth degree polynomial compensation	14.5726	2.3096	0.1142
Ninth degree polynomial compensation	14.7923	2.3076	−0.0344
10th degree polynomial compensation	14.7265	2.3053	0.0996

**Table 2 micromachines-13-01557-t002:** Comparison of the self-compensation effect of each group of data.

Type of Data Processing	Zero-Bias Variation (°/h)	Zero-Bias Stability (°/h)	Zero-Bias Mean (°/h)
Original data 1 (2 °C/min) Data 1 after compensation	67.68 5.40	27.8405 1.1256	- −0.1474
Original data 2 (2 °C/min) Data 2 after compensation	66.24 5.04	28.3720 1.1557	- 0.0298
Original data 3 (4 °C/min) Data 3 after compensation	65.88 6.48	27.8385 1.1943	- 0.1296
Original data 4 (4 °C/min) Data 4 after compensation	65.52 3.96	27.4579 0.8996	- −0.0535
Original data 5 (1 °C/min) Data 5 after compensation	62.28 6.48	23.5128 1.9715	- 0.0956

**Table 3 micromachines-13-01557-t003:** Determine the regression equation with data 1, and use the regression equation to compensate for the effect comparison of other data.

Type of Data Processing	Zero-Bias Variation (°/h)	Zero-Bias Stability (°/h)	Zero-Bias Mean (°/h)
Benchmark-data 1 (2 °C/min) Data 1 after compensation	66.24 5.04	28.3720 1.1557	- 0.0298
Original data 2 (1 °C/min) Data 2 after compensation	62.28 11.06	23.5128 3.3215	- −3.5240
Original data 3 (2 °C/min) Data 3 after compensation	67.6 87.92	27.8405 1.6827	- −1.3837
Original data 4 (4 °C/min) Data 4 after compensation	65.5 26.48	27.4579 1.5994	- −2.6233

**Table 4 micromachines-13-01557-t004:** Comparison of zero-bias drift compensation effects in the rapid heating-up stage of the gyroscope in the fast mode reversal.

Type of Data Processing	Zero-Bias Variation (°/h)	Zero Bias Stability (°/h) (1σ)	Zero-Bias Mean (°/h)
Mode reversal data 1 Data 1 after compensation	1.5424 0.3031	0.2890 0.0463	- $3.4141e^{-4}$
Mode reversal data 2 Data 2 after compensation	3.2637 0.5375	0.5184 0.0838	- $5.1325e^{-3}$
Mode reversal data 3 Data 3 after compensation	2.9499 0.4296	0.4678 0.0708	- $2.7213e^{-3}$
Mode reversal data 4 Data 4 after compensation	2.5438 0.4111	0.4009 0.0780	- $2.1863e^{-3}$

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
