# Peer review of "A Temperature Drift Suppression Method of Mode-Matched MEMS Gyroscope Based on a Combination of Mode Reversal and Multiple Regression"

_micromachines, 2022, doi:10.3390/mi13101557_

Round 1

Reviewer 1 Report

1. Mode reversal and real-time multiple regression compensation have been widely used to compensate the temperature drift of gyroscope bias. What's the novelty of this paper?

2.  How is the equation (5) derived from the equation (4)?

3. The multiple regression compensation model should be clearly presented, rather than giving the simplified model (8).

4. Figure 8 selects two groups of data under 2°C/min and three groups of data under 4°C/min. What is the reason for this choice?

Author Response

This document presents our full response to the Reviewers’ Comments – we have endeavored to make significant and best efforts in addressing all the technical issues raised. Our efforts include both extensive discussions and revision of the manuscript. We highly appreciate the feedbacks from the Reviewers; and we are very thankful to them for carefully reading our manuscript and for their recognition of the importance of this work, particularly their constructive comments and very helpful suggestions. Following these valuable advices, we have endeavored to respond to every point that the Reviewers suggested. Our detailed response is written in the attachment.

Reviewer 2 Report

The article presents a novel approach for MEMS gyroscope temperature drift suppression, based on the mode reversal and multiple regression algorithm. A sufficient background is provided in Introduction, with relevant cited references. The research design is appropriate and results are clearly presented.

However it should be noted that the algorithm of multiple regression is not aquately described. Most likely, it is based on the multiple mean square error technique and it would be better to describe it in detail. Only 2 formulas, (7) and (7) are not enough to understand the essence of the method.

In one of the equations of (4) one must change sign "-" for "+" to obtain Eq. (5).

Author Response

(The authors gave the same response as above.)

Round 2

Reviewer 1 Report

  • My comments have been solved.